# A RAF-SnRK2 kinase cascade mediates early osmotic stress signaling in higher plants

Zhen Lin[1,8], Yuan Li[2,3,8], Zhengjing Zhang[1,8], Xiaolei Liu[1,8], Chuan-Chih Hsu [1,7], Yanyan Du[1], Tian Sang[1], Chen Zhu[1], Yubei Wang[1], Viswanathan Satheesh [1], Pritu Pratibha [1], Yang Zhao [1], Chun-Peng Song [4], W. Andy Tao[5,6], Jian-Kang Zhu[1,3]* & Pengcheng Wang [1]*

Osmoregulation is important for plant growth, development and response to environmental changes. SNF1-related protein kinase 2s (SnRK2s) are quickly activated by osmotic stress and are central components in osmotic stress and abscisic acid (ABA) signaling pathways; however, the upstream components required for SnRK2 activation and early osmotic stress signaling are still unknown. Here, we report a critical role for B2, B3 and B4 subfamilies of Raf-like kinases (RAFs) in early osmotic stress as well as ABA signaling in *Arabidopsis thaliana*. B2, B3 and B4 RAFs are quickly activated by osmotic stress and are required for phosphorylation and activation of SnRK2s. Analyses of high-order mutants of *RAFs* reveal critical roles of the RAFs in osmotic stress tolerance and ABA responses as well as in growth and development. Our findings uncover a kinase cascade mediating osmoregulation in higher plants.

[1] Shanghai Center for Plant Stress Biology, CAS Center for Excellence in Molecular Plant Sciences, Chinese Academy of Sciences, Shanghai 200032, China. [2] State Key Laboratory of Plant Physiology and Biochemistry, College of Biological Sciences, China Agricultural University, Beijing 100193, China. [3] Department of Horticulture and Landscape Architecture, Purdue University, West Lafayette, IN 47907, USA. [4] Key laboratory of Plant Stress Biology, School of Life Sciences, Henan University, Kaifeng 475004, China. [5] Department of Biochemistry, Purdue University, West Lafayette, IN 47907, USA. [6] Department of Chemistry, Purdue University, West Lafayette, IN 47907, USA. [7] Present address: Department of Plant Biology, Carnegie Institution for Science, Stanford, CA 94305, USA. [8] These authors contributed equally: Zhen Lin, Yuan Li, Zhengjing Zhang, Xiaolei Liu. *email: jkzhu@sibs.ac.cn; pcwang@sibs.ac.cn

Drought, high salinity, and low temperatures cause osmotic stress to plants, greatly limiting plant productivity[1–3]. Osmoregulation is essential for plants to cope with environmental challenges and is also required for plant growth and development. SNF1-related protein kinase 2s (SnRK2s) are critical for osmotic stress responses[4,5]. Three of the ten SnRK2 family members, SnRK2.2, SnRK2.3, and SnRK2.6, are core components in the signaling pathway of abscisic acid (ABA)[6,7], a phytohormone that accumulates in plants subjected to osmotic stress[8–11]. In the absence of ABA, SnRK2.2/3/6 are inhibited by clade A protein phosphatase 2Cs (PP2Cs) through dephosphorylation[12–14]. Upon hyperosmotic stress, ABA accumulates and binds to its receptors, the PYRABACTIN RESISTANCE1 (PYR1)/PYR1-LIKE (PYL)/REGULATORY COMPONENTS OF ABA RECEPTORS (RCAR) family proteins, which subsequently inhibit PP2C activity, resulting in the release of SnRK2s from inhibition. Current models suggest that SnRK2.6 is self-activated by autophosphorylation after release from PP2C-mediated inhibition[15]. The activated SnRK2s phosphorylate downstream effectors to mediate stress responses[16,17].

All members of the SnRK2 family, except SnRK2.9, are also activated by osmotic stress[4]. The snrk2.1/2/3/4/5/6/7/8/9/10 decuple mutant, which lacks all ten members of the SnRK2 family, is hypersensitive to osmotic stress[5]. Neither the snrk2.2/3/6 triple nor snrk2.1/4/5/7/8/9/10 septuple mutant has an obvious osmotic stress-sensitive phenotype, suggesting redundancy among SnRK2s in the osmotic stress response[5]. Osmotic stress-mediated SnRK2 activation is independent of the ABA signaling pathway[4,5,18,19]. In ABA insensitive 1 (abi1-1), a dominant mutant where the ABI1 PP2C phosphatase cannot be inhibited by PYR/PYL/ACAR, or in high-order mutants of PYR/PYL/ACAR ABA receptors, osmotic stress-induced SnRK2 activation is not reduced[18,20], while ABA-induced SnRK2 activation is abolished[20,21]. How the SnRK2s are activated by osmotic stress is a major unanswered question.

Raf-like protein kinases (RAFs) have been classified as mitogen activated protein kinase kinase kinases (MAPKKKs) in plants[22,23]. According to sequence similarity, Raf-like protein kinases are classified into four B and seven C subgroups[22]. In the moss Physcomitrella patens, ABA and abiotic stress-responsive Raf-like Kinase (ARK), a B3 subfamily Raf-like kinase, participates in the regulation of both ABA and hyperosmotic stress responses by phosphorylating PpSnRK2[24]. The Arabidopsis thaliana genome contains 80 genes encoding Raf-like protein kinases, including four members of the B1 subgroup, six members of the B2 subgroup, six members of the B3 subgroup, and seven members of the B4 subgroup. One B4 subfamily member, Hydraulic Conductivity of Root 1 (HCR1), is involved in a potassium-dependent response to hypoxia[25]. In Arabidopsis, mutants of several members of the B2 and B3 families of Raf-like protein kinases such as ctr1, raf10 and raf11, are insensitive to ABA[26,27], and the sis8 mutant is hypersensitive to salt stress[28]. The phosphorylation of a B4 Raf-like protein kinase, AT1G16270, is up-regulated by mannitol treatment in Arabidopsis[29]. However, whether Raf-like protein kinases function in SnRK2 activation, and in osmotic stress and/or ABA signaling in higher plants remains unknown.

Here, we report a critical role for some Raf-like kinases in early osmotic stress as well as ABA signaling in Arabidopsis thaliana. B2, B3 and B4 RAFs are very quickly activated by osmotic stress and are required for phosphorylation and activation of SnRK2s. Analyses of high-order mutants of RAFs reveal critical roles of these RAFs in osmotic stress tolerance and ABA responses as well as in plant growth and development. Our findings uncover an upstream kinase cascade mediating osmoregulation and ABA signaling in higher plants.

## Results

**Osmotic stress activates protein kinase OKs.** To investigate the phosphorylation events in early osmotic stress signaling, we used in-gel kinase assays to measure kinase activation upon hyperosmotic stress caused by mannitol treatment[5,30]. Four groups of protein kinases were activated by mannitol treatment and ABA (Fig. 1a). SnRK2.2/3/6 (approximately 40 to 42 kDa) were strongly activated by both ABA and osmotic stress, while the ABA-independent SnRK2.1/4/5/9/10 (37 to 40 kDa) were strongly activated only by osmotic stress (Fig. 1b). In addition to the SnRK2s, we found two groups of protein kinases of approximately 130 and 100 kDa that were strongly activated by osmotic stress but not ABA (Fig. 1a). We termed these kinases osmotic stress-activated protein kinases (OKs). Strong activation of the 130-kDa OKs ($OK^{130}$) was observed at 2.5 min after mannitol treatment, peaking at 5 min (Fig. 1a). Activation of the 100-kDa OKs ($OK^{100}$) was clearly detectable after 5 min of mannitol treatment (Fig. 1a). Rapid OK activation in response to osmotic stress suggests a role for these kinases in early osmotic stress signaling. Activation of the OKs did not require SnRK2s and was independent of ABA signaling, since OK activation by mannitol treatment was still observed in the snrk2.2/3/6 triple (snrk2-triple) mutant, which is deficient in ABA signaling, and in snrk2.1/4/5/7/8/9/10 septuple

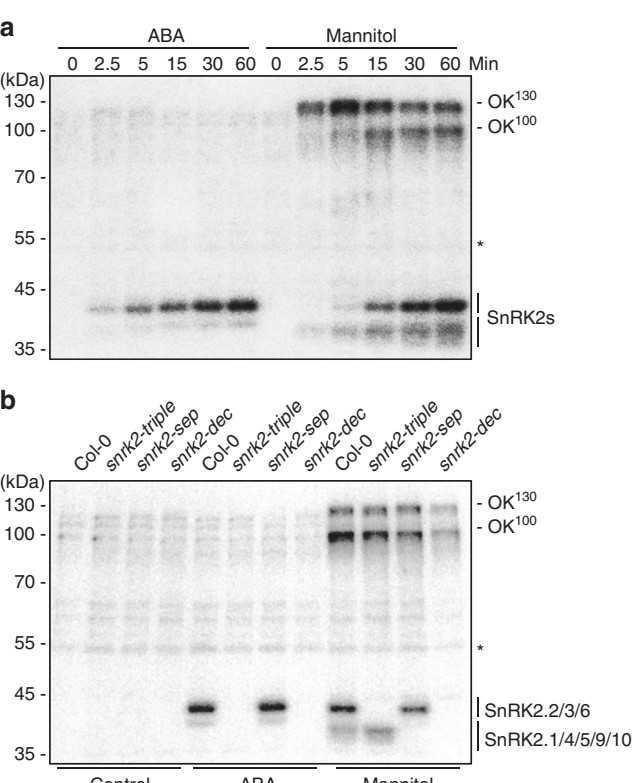

**Fig. 1 Osmotic stress activates OKs and SnRK2s. a** In-gel kinase assay showing the SnRK2, $OK^{100}$, and $OK^{130}$ activities after the indicated time of treatment with 50 μM ABA or 800 mM mannitol. **b** In-gel kinase assay showing the SnRK2, $OK^{100}$, and $OK^{130}$ activities in wild type, snrk2.2/3/6 triple (snrk2-triple), snrk2.1/4/5/7/8/9/10 septuple (snrk2-sep), and snrk2.1/2/3/4/5/6/7/8/9/10 decuple (snrk2-dec) mutants after 30 min of ABA or mannitol treatment. Though it is reported that SnRK2.7 and SnRK2.8 are also activated by ABA treatment, their activation could not be clearly indicated in our in-gel kinase assay. Asterisk indicates a non-induced band that could be used as a loading control. Images shown are representative of three independent experiments. Source data are provided as Source Data file.

(*snrk2-sep*) and *snrk2.1/2/3/4/5/6/7/8/9/10* decuple mutants (*snrk2-dec*)[5] (Fig. 1b).

**Identification of OKs by quantitative phosphoproteomics.** To determine the identity of the OKs, we used phosphoproteomic analysis to examine phosphoproteins in both the wild-type and *snrk2-dec* mutant plants after 30 min of mannitol treatment (Fig. 2a, Supplementary Fig. 1, Supplementary Data 1 to 5), since we expected that the OKs are autophosphorylated upon activation and that the autophosphorylation would occur in plants of both genotypes. Twenty-one phosphosites in 18 protein kinases were found to be up-regulated by mannitol treatment in both the wild-type and *snrk2-dec* mutant (Fig. 2a, Supplementary Data 4 and 5). These included several phosphosites in Raf-like protein kinases. B4 Raf-like kinases (117 to 140 kDa, see Fig. 2b) have an N-terminal PB1 domain and a C-terminal kinase domain[22]. Phosphopeptides from six of the seven B4 Raf-like kinases were significantly

up-regulated by osmotic stress, in both the wild-type and *snrk2-dec* mutant (Fig. 2c, see also Supplementary Data 4 and 5). Several phosphosites in members of the B2 and B3 subfamilies of the Raf-like kinases, RAF4 (AT1G18160), RAF5/Sugar Insensitive 8(SIS8), RAF2/Enhanced Disease Resistance 1(EDR1), RAF11, and RAF10, were also present in the list of mannitol-induced phosphoproteins (Fig. 2d, Supplementary Data 4). Members of the B2 and B3 groups have molecular weights from 75 to 112 kDa. Some of the phosphosites from the RAFs were located in the activation loop of these kinases. Phosphorylation in the activation loop is a conserved mechanism of protein kinase activation. Taking these results together, we hypothesized that members of the B4 Raf-like kinases may correspond to the OK[130], and that members of the B2 and B3 Raf-like kinases may be the OK[100].

**Mutational analyses identify the OKs as Raf-like kinases.** To validate our hypothesis, we first determined the activation of OKs

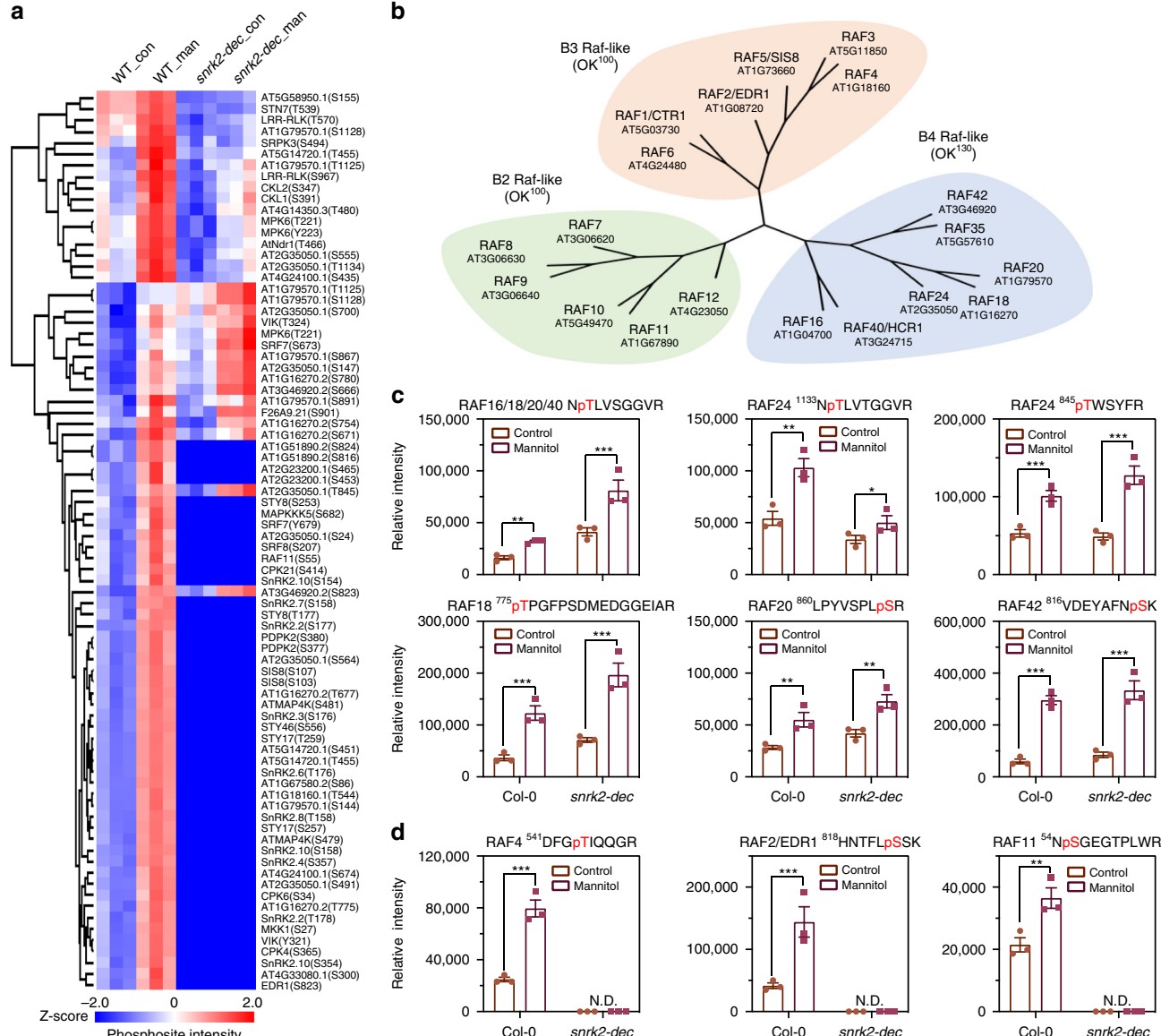

**Fig. 2 Identification of OKs by phosphoproteomics. a** Heat map showing relative intensity of phosphosites from protein kinases in control and mannitol-treated wild-type and *snrk2-dec* mutant seedlings. **b** Phylogenetic tree of B2, B3, and B4 RAFs. **c** Quantitative analysis of phosphopeptides from B4 Raf-like kinases in seedlings with or without mannitol treatment. Error bars, SEM ($n = 3$). Student's *t*-test, \**p* < 0.05, \*\**p* < 0.01, \*\*\**p* < 0.001. **d** Quantitative analysis of phosphopeptides from B2 and B3 Raf-like kinases in seedlings with or without mannitol treatment. Two-tailed paired *t*-tests, \*\**p* < 0.01, \*\*\**p* < 0.001. N.D., not identified in the samples. Source data are provided in Supplementary Data 4 and 5.

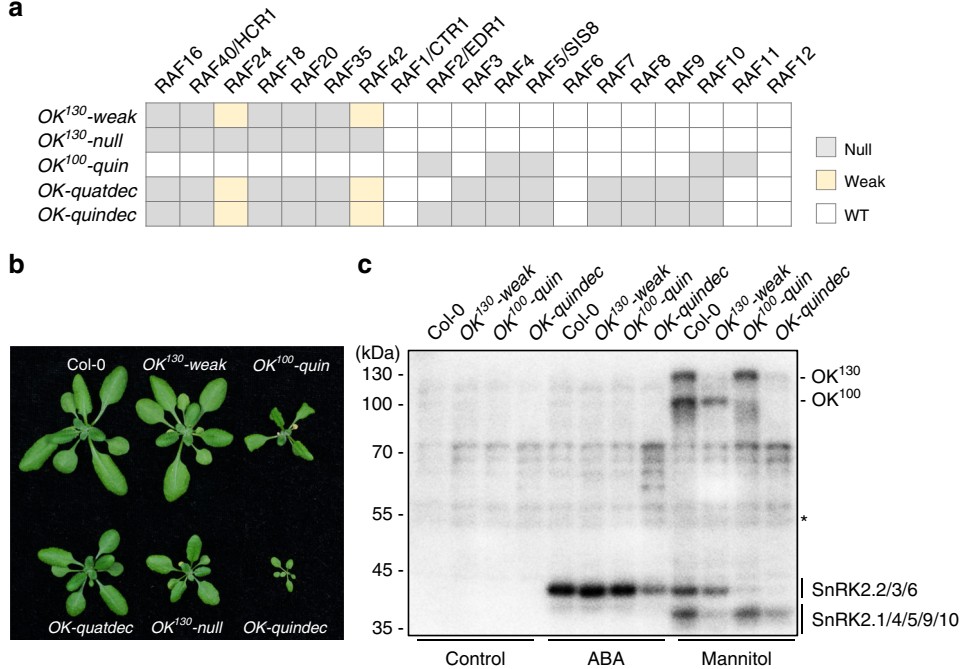

**Fig. 3 Genetic identification of OKs as subgroups of Raf-like protein kinases. a** Mutations of RAF genes in high-order mutants used in this study. Wild type (WT), knock-out mutant (null), or weak alleles (weak) are indicated with different colors. **b** Photographs of seedlings after 4 weeks of growth in the soil. **c** In-gel kinase assay showing the SnRK2 and OK activities after 15 min of ABA or mannitol treatments in wild-type and $OK^{130}$-weak, $OK^{100}$-quin, and $OK$-quindec mutants. Images shown are representative of two independent experiments. Source data are provided as Source Data file.

and SnRK2s in single mutants of the B4 Raf-like kinases. The overall signals of OKs in the single mutants *raf16*, *raf40/hcr1*, *raf24*, *raf18*, *raf20*, *raf35* and *raf42*, were comparable to those in the wild-type (Supplementary Fig. 2a). Using Clusters of Regularly Interspaced Short Palindromic Repeats/CRISPR-associated 9 (CRISPR/Cas9)-mediated genome editing in a T-DNA insertion mutant line *raf16* (*salk_007884*), we generated a high-order mutant, $OK^{130}$-weak, containing frameshift/early termination mutations in *raf40/hcr1*, *raf18*, *raf20* and *raf35*, and 30 and 18 bp deletions in *raf24* and *raf42*, respectively (Fig. 3a, Supplementary Fig. 2b, c). $OK^{130}$ activation was strongly, although not completely, impaired in the $OK^{130}$-weak mutant (Supplementary Fig. 2d, f), supporting our hypothesis that the B4 subfamily represents $OK^{130}$. The mannitol-induced activation of SnRK2.1/4/5/9/10 was markedly, but not completely abolished in $OK^{130}$-weak (Fig. 3c), suggesting that the activation of SnRK2.1/4/5/9/10 is dependent on $OK^{130}$. $OK^{100}$ activation was also weakened or delayed in the $OK^{130}$-weak mutant, when compared to that in the wild-type (Supplementary Fig. 2d).

To eliminate the remaining $OK^{130}$ activity, we introduced additional mutations into $OK^{130}$-weak using a second CRISPR/Cas9 construct containing additional guide RNAs targeting the 5' regions of *raf24* and *raf42* and isolated an $OK^{130}$-null mutant with frameshift/null mutations in all seven members of the subfamily (Fig. 3a and Supplementary Fig. 2e). As expected, osmotic stress-activation of $OK^{130}$ and SnRK2.1/4/5/9/10 was completely abolished in the $OK^{130}$-null mutant (Supplementary Fig. 2f). These results further support that the B4 subfamily Raf-like kinases correspond to the $OK^{130}$, and show that the $OK^{130}$ genes redundantly control the activation of SnRK2.1/4/5/9/10 upon osmotic stress. By contrast, the activation of SnRK2.2/3/6 by ABA and osmotic stress in the $OK^{130}$-null mutant was comparable to that in the wild-type (Supplementary Fig. 2f), and the activation of the $OK^{100}$ upon osmotic stress was only partially impaired in the $OK^{130}$-null mutant (Supplementary Fig. 2f). This suggests that

osmotic stress-induced activation of $OK^{100}$ and SnRK2.2/3/6 is not dependent on $OK^{130}$.

To further identify the $OK^{100}$ and study how osmotic stress activates SnRK2.2, SnRK2.3 and SnRK2.6, we generated high-order mutants by introducing mutations into B2 and B3 subfamily members in wild-type and $OK^{130}$ mutant backgrounds (Fig. 3a). As $OK^{130}$-null plants produce few seeds (Supplementary Fig. 2g, h), we had to use the $OK^{130}$-weak plants to generate higher order mutants. Gene editing in the wild-type background resulted in $OK^{100}$-quin (*raf2/edr1;raf4;raf5/sis8;raf10;raf11*) (Fig. 3a, Supplementary Fig. 3a, b). The phosphopeptides from the five protein kinases showed mannitol up-regulation (Fig. 2a). Gene editing in the $OK^{130}$-weak background produced an OK-quatdec mutant (*raf16;raf40;RAF24^{Δ10};raf18;raf20;raf35;RAF42^{Δ6};raf3;raf4;raf5/sis8;raf7;raf8;raf9;raf10*) and an OK-quindec mutant (*raf16;raf40;RAF24^{Δ10};raf18;raf20;raf35;RAF42^{Δ6};raf2/edr1;raf3;raf4;raf5/sis8;raf7;raf8;raf9;raf10*) (Fig. 3a and Supplementary Fig. 3a, 3c, d). $OK^{100}$-quin showed strong growth inhibition phenotypes and OK-quindec showed extremely arrested growth (Fig. 3b, Supplementary Fig. 4a). OK-quatdec, which differed from OK-quindec by having wild-type RAF2/EDR1, showed only a slightly inhibited growth (Fig. 3b). The mannitol-triggered activation of the $OK^{100}$ and SnRK2.2/3/6 was almost completely abolished in the OK-quindec mutant (Fig. 3c). ABA still activated SnRK2.2/3/6 in the OK-quindec seedlings, but the activation was much weaker than that in the wild type (Fig. 3c). Together, our results show that B4 RAFs correspond to the $OK^{130}$, and that members of the B2 and B3 Raf-like kinases are the $OK^{100}$.

We examined the expression patterns of *RAFs* in different tissues and stress conditions from an online eFP Browser database (http://bar.utoronto.ca/efp2). The various *RAFs* were expressed in different tissues and some *RAFs* were highly expressed in dry seeds and mature pollens (Supplementary Fig. 4b), consistent with the observation that $OK^{130}$-null produces fewer seeds than the wild type (Supplementary Fig. 2g, h). Interestingly, the expression of

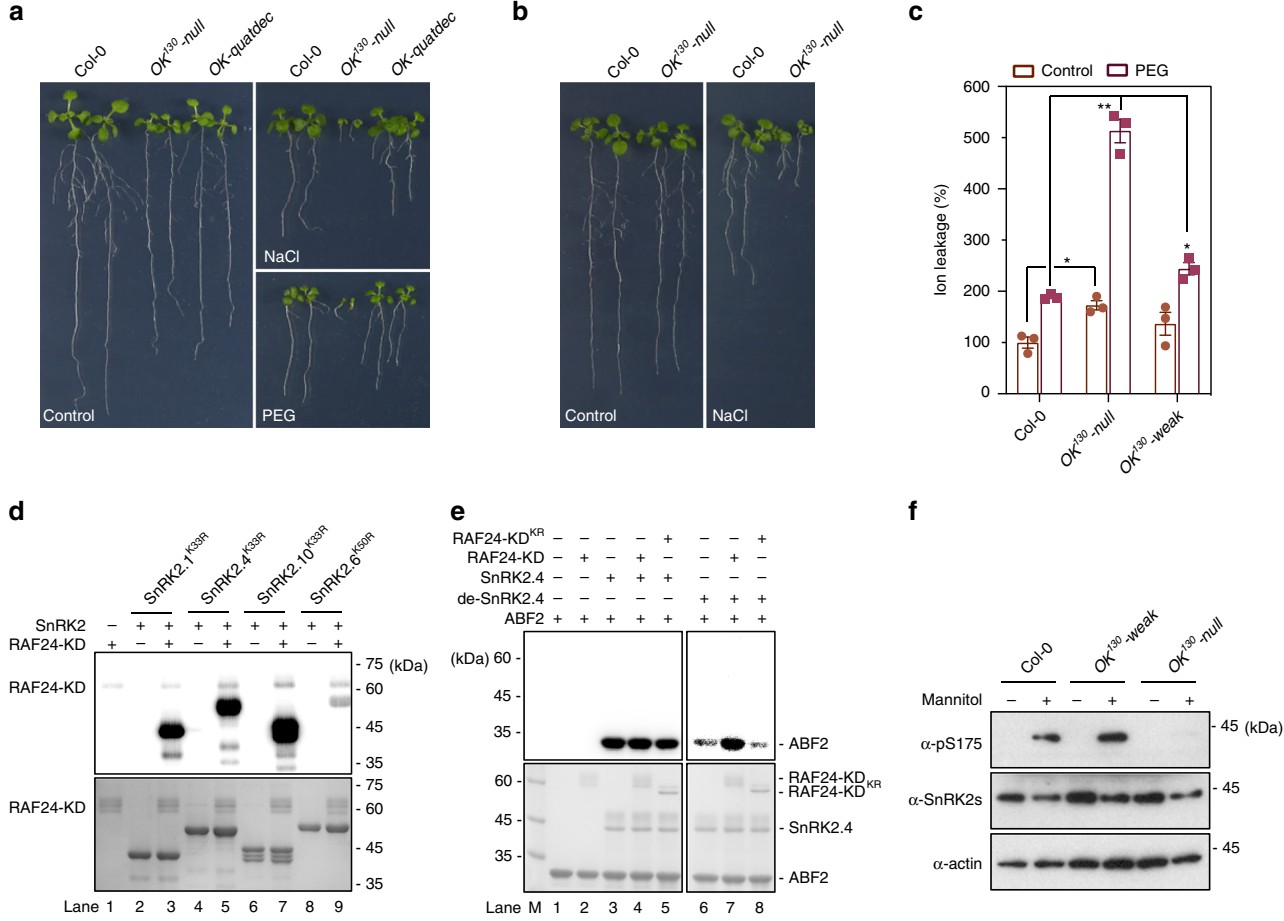

**Fig. 4 RAF deficiency alters osmotic stress sensitivity. a** Photographs of seeds after 10 days of germination and growth on 1/2 Murashige and Skoog (MS) medium containing 100 mM NaCl or −0.5 MPa PEG. **b** Photographs of seedlings 7 days after transfer to and growth on 1/2 MS medium containing 100 mM NaCl. **c** Osmotic stress sensitivity as measured by electrolyte leakage in seedlings treated with 30% PEG. Error bars, SEM ($n = 3$). Two-tailed paired $t$-tests, *$p < 0.05$, **$p < 0.01$. **d** RAF24-KD phosphorylates SnRK2s in vitro. Recombinant RAF24-KD was used to phosphorylate SnRK2s expressed and purified from *E. coli* in the presence of [γ-$^{32}$p]ATP. Autoradiograph (upper) and Coomassie staining (lower) show phosphorylation and loading of purified RAF24-KD and SnRK2s, respectively. **e** RAF24 activates dephosphorylated SnRK2.4 in vitro. RAF24-KD enhances the activity of pre-dephosphorylated SnRK2.4 (de-SnRK2.4) (right panel), but not the activity of SnRK2.4 without pre-dephosphorylation (left panel). Phosphorylation of ABF2 fragment was used to indicate the SnRK2.4 activity. Autoradiograph (upper) and Coomassie staining (lower) show phosphorylation and loading of purified RAF24-KD and SnRK2.4, respectively. **f** The mannitol-induced phosphorylation of the conserved serine corresponding to Ser175 in SnRK2.6 depends on RAFs. Source data are provided as Source Data file.

some *RAFs*, especially in the root, was highly induced by osmotic stress caused by mannitol and high salt (Supplementary Fig. 4c). ABA also up-regulated the expression of *RAF35*, *RAF6* and *RAF12* (Supplementary Fig. 4d). By generating transgenic lines expressing GFP fusions, we examined the subcellular localization of RAF proteins. GFP-RAF20, GFP-RAF12 and GFP-RAF7 were localized to the cytosol, while RAF11 was localized to small spots in the cytosol (Supplementary Fig. 5).

**RAFs are required for osmotic stress tolerance**. We examined the responses of high-order mutants of *RAFs* to osmotic stress and found that $OK^{130}$-*null* but not $OK^{130}$-*weak* was hypersensitive to osmotic stress caused by mannitol, NaCl, or polyethylene glycol (PEG) treatment, in assays of seed germination, seedling growth, and electrolyte leakage (Fig. 4a–c and Supplementary Fig. 6c, d). *OK-quatdec*, containing weak alleles of *RAF24* and *RAF42*, showed only mild hypersensitivity to osmotic stresses (Fig. 4a–c and Supplementary Fig. 6a, b), consistent with the mild osmotic stress sensitivity of $OK^{130}$-*weak* (Supplementary Fig. 6c, d).

Upon ABA treatment, the activation of SnRK2.6 depends on the inhibition of the A clade PP2C by ABA-PYLs[12,21,31]. Consistent with this notion, ABA-induced SnRK2.2, SnRK2.3 and SnRK2.6 activation was enhanced in the *abi1/abi2/pp2ca* triple mutant, but strongly impaired in *abi1-1* (Supplementary Fig. 7a). By contrast, the osmotic stress-induced activation of SnRK2s was not, or only slightly, affected by mutations in the PP2Cs (Supplementary Fig. 7a). The notion that osmotic stress-triggered activation of SnRK2s does not require ABA signaling is supported by our previous work[20], as well as studies from other groups[18,32]. Interestingly, the activation of both $OK^{130}$ and $OK^{100}$ by osmotic stress is also not impaired in the *abi1-1* mutant (Supplementary Fig. 7a), suggesting that the activation is not due to inhibition of the clade A PP2Cs.

**RAFs interact with and phosphorylate SnRK2s**. Our findings that B4 Raf-like kinases were activated earlier than SnRK2s and were required for the SnRK2 activation suggested that the B4 Raf-like kinases may directly activate SnRK2s by phosphorylation. To test this hypothesis, we first used immunoprecipitation-mass

spectrometry (IP-MS) to detect possible interactions between SnRK2s and B4 Raf-like kinases in plants (Supplementary Fig. 7b). We found several peptides from SnRK2.1/2/4/5/9/10 in anti-GFP immunoprecipitates from GFP-RAF40, GFP-RAF20, or GFP-RAF35, but not from empty GFP plants, implying that the B4 Raf-like kinases and SnRK2s are associated in vivo in plants. A split luciferase complementation assay on RAF35 and several SnRK2s supported the association and suggested direct physical interactions between RAF35 and the tested SnRK2s (Supplementary Fig. 7c).

We then tested whether recombinant B4 Raf-like kinase proteins may phosphorylate SnRK2s. The recombinant kinase domains (KDs) of RAF40/HCR1 and RAF24 displayed detectable kinase activities in vitro. Recombinant RAF24-KD strongly phosphorylated the full-length SnRK2.1$^{K33R}$, SnRK2.4$^{K33R}$, and SnRK2.10$^{K33R}$, the "kinase-dead" mutant versions of the osmotic stress-activated SnRK2s lacking autophosphorylation (Fig. 4d, lane 3, 5, and 7). The RAF24-KD also weakly phosphorylated SnRK2.6$^{K50R}$, a "kinase-dead" mutant of the ABA-activated SnRK2.6 (Fig. 4d, lane 9). Similarly, RAF40/HCR1-KD strongly phosphorylated SnRK2.4$^{K33R}$ and SnRK2.10$^{K33R}$, and weakly phosphorylated SnRK2.6$^{K50R}$ (Supplementary Fig. 7d).

To identify the RAF target sites in the SnRK2s, we used mass spectrometry to identify phosphopeptides from the above in vitro kinase reactions. We identified 23 and 6 phosphopeptides from SnRK2.4$^{K33R}$ and SnRK2.6$^{K50R}$, respectively, after in vitro kinase reactions with different B4 Raf-like kinases using $^{18}$O-ATP as the phosphate donor (Supplementary Fig. 8, Supplementary Data 6). Six putative RAF target sites, Ser158, Ser162, Ser166, Thr167, Thr170 and Ser180, were found in phosphopeptides coming from the activation loop of SnRK2.4 (Supplementary Fig. 8a–d, Supplementary Data 6), and two putative target sites, Ser171 and Ser175, were found in the same region in SnRK2.6 (Supplementary Fig. 8e, Supplementary Data 6). The phosphorylation of a highly conserved site in SnRK2s, corresponding to Ser175 in SnRK2.6, is essential for SnRK2.6 activation[31]. Another conserved site corresponding to Ser171 in SnRK2.6 is also crucial for osmotic stress- and ABA-mediated SnRK2 activation[18]. Ser to Ala mutations of these sites partially reduced but did not abolish the phosphorylation signal in the in vitro kinase assay (Supplementary Fig. 8f), which is consistent with our mass spectrometry results showing that multiple sites in addition to the activation loop residues in SnRK2.4 and SnRK2.6 are phosphorylated by Raf-like kinases (Supplementary Data 6).

Since phosphor-mimicking and non-phosphorylatable mutations in the activation loop render SnRK2 inactive[28], we could not evaluate the contribution of these sites to SnRK2 activation by directly mutating them. So, we performed an in vitro kinase assay with the phosphorylation of an ABA-responsive element-Binding Factor 2(ABF2) fragment, a well-defined SnRK2 substrate[12,30], as an indicator of SnRK2 activity. Recombinant RAF24-KD itself did not phosphorylate ABF2 (Fig. 4e, lane 2). Adding RAF24-KD did not enhance the existing activity of SnRK2.4 (Fig. 4e, lane 4), which might be because the recombinant SnRK2.4 was already highly auto-phosphorylated and fully activated in E. coli. After being dephosphorylated by ABI1 in vitro, the full-length SnRK2.4 showed very weak kinase activity (Fig. 4e, lane 6). This suggests that dephosphorylated SnRK2.4 has no or only very weak self-activation activity, which contrasts with the previous hypothesis that SnRK2s have strong auto-phosphorylation activity and self-activate when they are not inhibited by PP2C[15,31]. Interestingly, co-incubating with RAF24-KD, but not with the "kinase-dead" form RAF24-KD$^{K1001R}$, substantially increased the kinase activity of dephosphorylated-SnRK2.4 (Fig. 4e, lane 7 and 8). Finally, by immunoblotting using an anti-SnRK2.6-pS175 antibody[20], which recognizes the phosphorylated serine residue in the

activation loop of multiple SnRK2s corresponding to Ser175 in SnRK2.6, we found that the Ser175 phosphorylation triggered by osmotic stress was abolished in OK$^{130}$-null seedlings (Fig. 4f). Together with the in-gel kinase assay result (Fig. 3c), our findings suggest that the phosphorylation of SnRK2s, especially the conserved serine residue in the activation loop, by Raf-like kinases is required for osmotic stress-triggered SnRK2 activation.

**RAFs are required for ABA-mediated SnRK2 activation**. The strong reduction in ABA-triggered SnRK2 activation in OK-quindec (Fig. 3c) suggested that Raf-like kinases may regulate ABA responses. Since the OK-quindec plants produced very few seeds for subsequent experiments, we tested ABA responses in OK-quatdec mutant plants and found that the mutant was insensitive to ABA during seed germination and post-germination seedling growth (Fig. 5a, b, and Supplementary Fig. 9a, b). In addition, OK-quatdec mutant seedlings showed higher water loss than the wild type and the other tested RAF high-order mutants, phenocopying the snrk2-triple mutant (Fig. 5c). These results suggest that the RAFs also participate in ABA-triggered SnRK2.2/3/6 activation. This notion is supported by the observation that all five tested kinase domains of B2 and B3 RAFs, RAF5/SIS8-KD, RAF2/EDR1-KD, RAF6-KD, RAF10-KD and RAF7-KD, could phosphorylate SnRK2.6$^{K50R}$ (Supplementary Fig. 9c). Interestingly, RAF6-KD and RAF10-KD showed a stronger capability to phosphorylate ABA-dependent SnRK2s (SnRK2.6 and SnRK2.8 in our assay) than ABA-independent SnRK2s (SnRK2.1, SnRK2.4, and SnRK2.10 in the assay) (Fig. 5d and Supplementary Fig. 9d). A yeast-two-hybrid assay showed that B2 and B3 subgroup RAFs interact with SnRK2.6 but not with SnRK2.4 (Supplementary Fig. 9e). We also found that, like SnRK2.4, the dephosphorylated-SnRK2.6 (i.e., pretreated with ABI1) was incapable of self-activation (Fig. 5e, lane 7). However, adding RAF5-KD, RAF6-KD, or RAF10-KD strongly increased the phosphorylation of the dephosphorylated SnRK2.6 and therefore increased the phosphorylation of ABF2 (Fig. 5e, lane 8–10). RAF40-KD and RAF24-KD had almost no effect on the phosphorylation of dephosphorylated-SnRK2.6 and ABF2 (Fig. 5e, lane 11, 12), further indicating specificity between subgroups of RAF-like kinases and SnRK2s. Consistent with this, the ABA-induced phosphorylation of Ser175 was only abolished in the OK$^{130}$-quatdec mutant but not in the OK$^{130}$-weak or OK$^{130}$-null allele (Supplementary Fig. 9f).

Consistent with the strong ABA-insensitive phenotypes of OK-quatdec mutant plants, the osmotic stress- and ABA-induced transcript accumulation of several ABA-responsive genes, like Responsive to Desiccation 29B (RD29B), Responsive to ABA 18 (RAB18), and Cold-Regulated 15 A (COR15A), was dramatically impaired in the OK-quatdec mutant (Fig. 5f). The expression of some ABA-responsive transcription factors, e.g., ABF2 and ABF4, was also partially impaired in the OK-quatdec mutant when compared to the wild type (Supplementary Fig. 9g).

## Discussion

Our results show that the B2, B3, and B4 subfamilies of Raf-like protein kinases are upstream kinases that phosphorylate and activate SnRK2s and are critical in mediating osmotic stress and ABA responses. The RAFs are likely also important for osmoregulation during growth and development, as the OK$^{130}$-null and OK-quindec mutants show strong growth and developmental defects. The plant RAFs are presumed to be MAPKKKs[22], although their ability to phosphorylate MAPKKs in plants has not been characterized biochemically or genetically. Our results suggest that the 19 group B Raf-like protein kinases together with 10 SnRK2s form a kinase cascade in early osmotic stress and ABA signaling. The OK$^{130}$/B4 Raf-like kinases prefer to phosphorylate

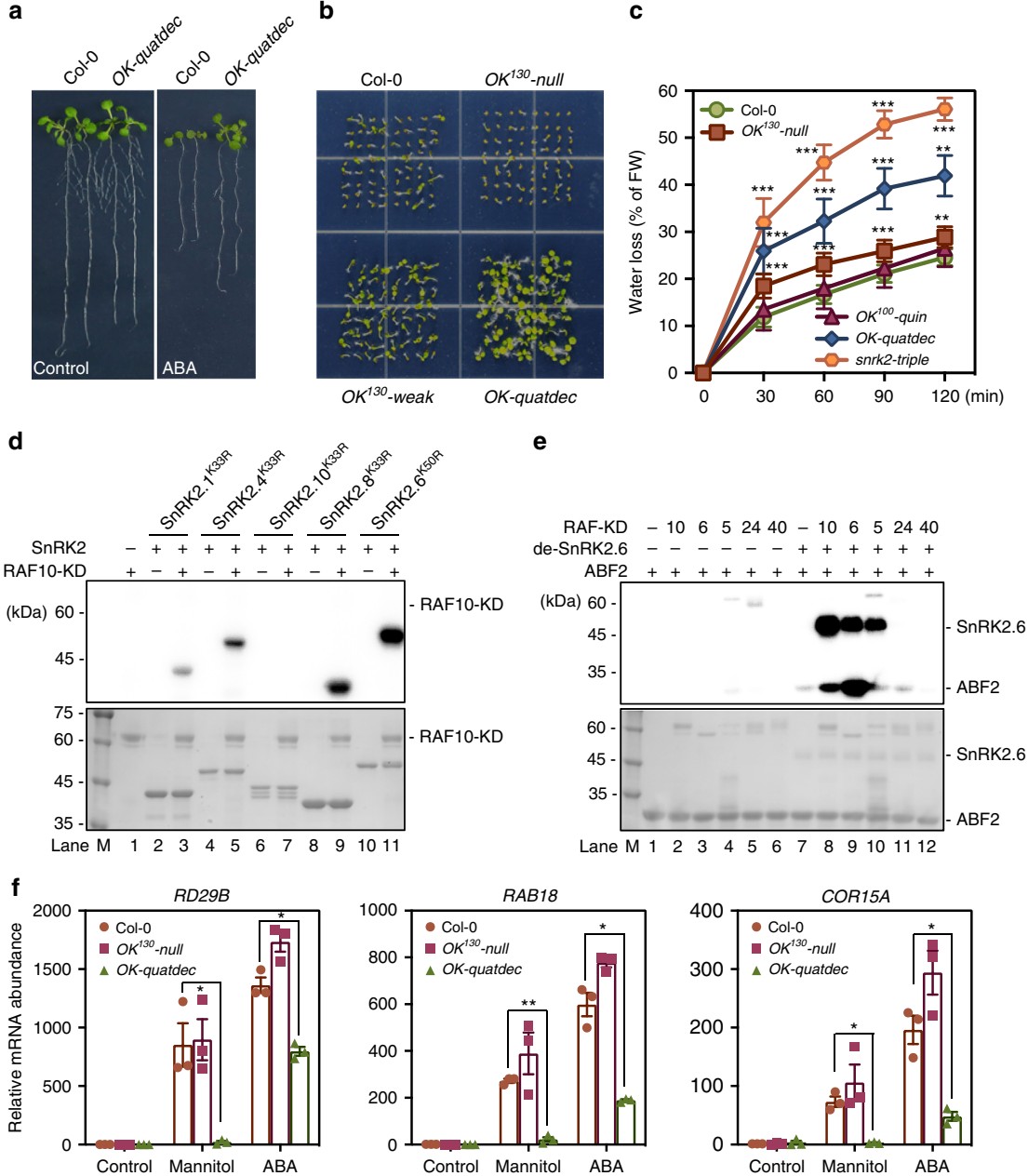

**Fig. 5 B2 and B3 Raf-like kinases mediate ABA signaling by phosphorylating SnRK2s. a** Photographs of seedlings 7 days after transfer to and growth on 1/2 MS medium containing 20 μM ABA. **b** Photographs of seeds after 7 days of germination on 1/2 MS medium containing 1 μM ABA. **c** Water loss of the 4-week-old wild type and high-order RAF mutants. Error bars, SD ($n = 5$). Two-tailed paired $t$-tests, *$p < 0.05$, **$p < 0.01$, ***$p < 0.001$. **d** RAF10-KD phosphorylates SnRK2s in vitro. Autoradiograph (upper) and Coomassie staining (bottom) show phosphorylation and loading of purified RAF10-KD and SnRK2s, respectively. **e** Adding RAF5-KD, RAF6-KD or RAF10-KD, but not RAF40-KD or RAF24-KD, strongly increases the phosphorylation of pre-dephosphorylated SnRK2.6 and ABF2 fragment. Autoradiograph (upper) and Coomassie staining (lower) show phosphorylation and loading of purified RAF-KD and SnRK2.6, respectively. **f** Expression of stress-responsive genes in wild type, $OK^{130}$-null, and OK-quatdec seedlings after 6 h of mannitol or ABA treatments. Error bars, SEM ($n = 3$). Two-tailed paired $t$-tests, *$p < 0.05$, **$p < 0.01$. Source data are provided as Source Data file.

ABA-independent SnRK2s, while OK$^{100}$/B2&3 Raf-like kinases favor phosphorylation of ABA-dependent SnRK2s (Figs. 3c, 4d, 5d, Supplementary Figs. 7d, 9c, d). Our discovery of the group B Raf-like kinases as the upstream kinases for osmotic stress-triggered activation of SnRK2s advances our understanding of osmoregulation in plants. In addition, our results suggest that transphosphorylation of the ABA-dependent SnRK2s by group B2 and B3 Raf-like kinases is a pre-requisite for the activation of SnRK2s in ABA signaling, which provides an important update on our current understanding of the ABA core signaling

pathway. Since SnRK2.6 purified from *E. coli* can autophosphorylate and can be activated in vitro by ABA in a test tube reconstitution of the core ABA signaling pathway with PYR/PYL/ACAR and clade A PP2C[12,15,30], it has been assumed that SnRK2.6 autophosphorylation is sufficient for its activation. Our results here with dephosphorylated SnRK2.6 show that transphosphorylation by group B Raf-like kinases is necessary, suggesting that previous in vitro assay results were affected by some kinase(s) in *E. coli* that can transphosphorylate SnRK2.6.

Although group B2 and B3 Raf-like kinases are required for SnRK2.2/3/6 activation in both ABA and osmotic stress, the mechanisms of SnRK2 activation might differ. ABA-induced SnRK2 activation is impaired in the *abi1-1* mutant, whereas osmotic stress-induced SnRK2.2/3/6 activation is not affected (Supplementary Fig. 7a). We noticed that RAFs phosphorylate SnRK2.6 on both Ser171 and Ser175 (Supplementary Fig. 8), which are known as direct target sites of the PP2C phosphatases[31]. Additional RAF phosphosites exist in SnRK2.6 besides Ser171 and Ser175 (Supplementary Data 6). The phosphorylation of these additional phosphosites may circumvent the PP2C-mediated inhibition to cause activation of the SnRK2s under osmotic stress. Further studies are needed to determine the detailed biochemical mechanisms that differentiate the SnRK2 activation by ABA and osmotic stress in plants.

Due to the large number of kinases in the RAF-SnRK2 cascade and the functional redundancy between the members, it will be challenging to dissect the unique functions of each RAFs. Furthermore, we suspect coordination between the RAFs and feedback regulation of the RAFs by their downstream SnRK2s in the cascade, given that subgroup I SnRK2 activation was enhanced in the *OK-quindec* mutant compared to the *OK130-weak* mutant (Fig. 3c), and that OK130 and OK100 activation was reduced in *snrk2-dec* (Fig. 1b).

The upstream component(s) that senses hyperosmolarity and quickly activates RAFs still needs to be identified. The activation of RAFs or SnRK2s by mannitol or sorbitol was not altered in a septuple mutant of reduced hyperosmolality induced $[Ca^{2+}]_i$ increase (*osca*) (Supplementary Fig. 10). OSCA1 is a putative osmosensor required for sorbitol-induced $Ca^{2+}$ signaling[33]. Our results suggest that activation of RAFs and SnRK2s is independent of OSCA and OSCA-mediated $Ca^{2+}$ signaling.

The regulation of SnRK2s by the B3 subfamily of Raf-like kinases appears to be conserved in land plants. ARK, the sole B3 Raf-like kinase in the moss *Physcomitrella patens*, also participates in regulation of ABA and hyperosmotic stress responses by phosphorylating PpSnRK2[24]. The identification of the RAF-SnRK2 cascade advances our understanding of osmotic stress and ABA signaling in higher plants, and provides potential molecular targets for engineering crops resilient to harsh environments.

## Methods

**Germination or growth under osmotic stress treatment**. Seeds were surface-sterilized for 10 min in 70% ethyl alcohol, and then rinsed four times in sterile-deionized water. For germination assays, sterilized seeds were grown on medium containing 1/2 MS nutrients, pH 5.7, with or without the indicated concentration of ABA, mannitol, NaCl or PEG, and kept at 4 °C for 3 days. Radicle emergence was analyzed 72 h after placing the plates at 23 °C under a 16 h light/8 h dark photoperiod. Photographs of seedlings were taken at indicated times after transfer to light. For growth assays, sterilized seeds were grown vertically on 0.85% agar containing 1/2 MS, pH 5.7, and kept at 4 °C for 3 days. Seedlings were grown vertically for 3–4 days and then transferred to medium with or without the indicated concentration of ABA, NaCl, mannitol or PEG. Root length was measured at the indicated days.

**In-gel kinase assay**. For in-gel kinase assays, 20 μg extract of total proteins was used for SDS/PAGE analysis with histone embedded in the gel matrix as the kinase substrate. After electrophoresis, the gel was washed three times at room temperature with washing buffer (25 mM Tris-Cl, pH 7.5, 0.5 mM DTT, 10 mM $Na_3VO_4$, 5 mM NaF, 0.5 mg/mL BSA, and 0.1% Triton X-100) and incubated at 4 °C overnight with three changes of renaturing buffer (25 mM Tris-HCl, pH 7.5, 1 mM DTT, 0.1 mM $Na_3VO_4$, and 5 mM NaF). The gel was then incubated at room temperature in 30 mL reaction buffer (25 mM Tris-Cl, pH 7.5, 2 mM EGTA, 12 mM $MgCl_2$, 1 mM DTT, and 0.1 mM $Na_3VO_4$) with 200 nM ATP plus 50 μCi of $[\gamma-^{32}P]ATP$ for 90 min. The reaction was stopped by transferring the gel into 5% (w/w) trichloroacetic acid and 1% (w/w) sodium pyrophosphate. The gel was washed in the same solution for at least 5 h with five changes of the wash solution. Radioactivity was detected with a Personal Molecular Imager (Bio-Rad).

**Protein extraction and digestion**. Protein extraction and digestion was performed as previously described[30]. Plants were lysed in lysis buffer (6 M guanidine hydrochloride in 100 mM Tris-HCl, pH 8.5) with 10 mM NaF, EDTA-free protease, and phosphatase inhibitor cocktails (Sigma-Aldrich, St. Louis, MO). Disulfide bonds in proteins were reduced and alkylated with 10 mM Tris(2-carboxyethyl)phosphine hydrochloride and 40 mM 2-chloroacetamide at 95 °C for 5 min. Protein lysate was precipitated using the methanol-chloroform precipitation method. Precipitated protein pellets were suspended in digestion buffer (12 mM sodium deoxycholate and 12 mM sodium lauroyl sarcosinate in 100 mM Tris-HCl, pH 8.5) and then were 5-fold diluted with 50 mM TEAB buffer. Protein amount was quantified using the BCA assay (Thermo Fisher Scientific, Waltham, MA). One mg of protein was then digested with Lys-C (Wako, Japan) in a 1:100 (v/w) enzyme-to-protein ratio for 3 h at 37 °C, and trypsin (Sigma-Aldrich, St. Louis, MO) was added to a final 1:100 (w/w) enzyme-to-protein ratio overnight. The detergents were separated from digested peptides by acidifying the solution using 10% TFA and then centrifuged at 16,000 g for 20 min. The digests were then desalted using a 100 mg SEP-PAK C18 cartridge (Waters, Milford, MA).

**Tandem mass tag (TMT) labeling**. The tryptic peptides (400 μg) from each replicate were dissolved in 100 μL of 200 mM HEPES (pH 8.5) and incubated with ACN-dissolved 0.8 mg of TMT 6-plex reagent (Thermo Fisher Scientific, Waltham, MA) for 1 h at room temperature. The reaction was quenched by adding 8 μL of 5% hydroxylamine to the sample and incubating for 15 min. All samples labeled with each TMT channel were pooled in a new tube, and the final concentration of ACN was diluted to less than 5% before desalting. The labeled phosphopeptides were desalted using a SEP-PAK C18 cartridge.

**Phosphopeptide enrichment**. Phosphopeptide enrichment was performed according to the reported IMAC StageTip protocol with some modifications[34,35]. The in-house-constructed IMAC tip was made by capping the end with a 20 μm polypropylene frits disk (Agilent). The tip was packed with 5 mg of Ni-NTA silica resin by centrifugation at 200 g for 1 min. $Ni^{2+}$ ions were removed by 100 μL of 100 mM EDTA solution. The tip was then activated with 100 μL of 100 mM $FeCl_3$ and equilibrated with 100 μL of loading buffer (6% (v/v) acetic acid at pH 3.0) prior to sample loading. The TMT-labeled peptides (400 μg) were reconstituted in 100 μL of loading buffer and loaded onto the IMAC tip. After successive washes with 200 μL of washing buffer (4% (v/v) TFA, 25% acetonitrile (ACN)) and 100 μL of loading buffer, the bound phosphopeptides were eluted with 150 μL of 200 mM $NH_4H_2PO_4$. The eluted phosphopeptides were loaded into a C18 beads StageTip and separated into eight fractions using high-pH reverse phase fractionation. The fractionated phosphopeptides were dried using a SpeedVac.

**LC-MS/MS analysis**. The phosphopeptides were dissolved in 5 μL of 0.25% formic acid (FA) and injected into an Easy-nLC 1000 (Thermo Fisher Scientific). Peptides were separated on a 45 cm in-house packed column (360 μm OD × 75 μm ID) containing C18 resin (2.2 μm, 100 Å, Michrom Bioresources). The mobile phase buffer consisted of 0.1% FA in ultra-pure water (Buffer A) with an eluting buffer of 0.1% FA in 80% ACN (Buffer B) run over a linear 90 min gradient of 6–30% buffer B at flow rate of 250 nL/min. The Easy-nLC 1000 was coupled online with a Velos Pro LTQ-Orbitrap mass spectrometer (Thermo Fisher Scientific). The mass spectrometer was operated in the data-dependent mode in which a full-scan MS (from $m/z$ 350–1500 with the resolution of 60,000 at $m/z$ 400) was followed by top 10 higher-energy collision dissociation (HCD) MS/MS scans of the most abundant ions with dynamic exclusion for 60 s and exclusion list of 500. The normalized collision energy applied for HCD was 40% for 10 ms activation time.

**Proteomics data search**. The raw files were searched directly against the *Arabidopsis thaliana* database (TAIR10 with 35,386 entries) with no redundant entries using MaxQuant software (version 1.5.4.1) with reporter ion MS2 type. Peptide precursor mass tolerance was set at 20 ppm, and MS/MS tolerance was set at 20 ppm. Search criteria included a static carbamidomethylation of cysteines (+57.0214 Da) and variable modifications of (1) oxidation (+15.9949 Da) on methionine residues, (2) acetylation (+42.011 Da) at N-terminus of protein, and (3) phosphorylation (+79.996 Da) on serine, threonine or tyrosine residues were searched. Search was performed with full tryptic digestion and allowed a maximum of two missed cleavages on the peptides analyzed from the sequence database. The false discovery rates of proteins, peptides and phosphosites were set at 1% FDR. The minimum peptide length was six amino acids, and a minimum Andromeda score was set at 40 for modified peptides. The phosphorylation sites induced by mannitol treatment in Col-0 and *snrk2-dec* mutant plants were selected using Perseus software (version 1.6.2.1). The intensities of phosphorylation sites were log2 transformed, and the quantifiable phosphorylation sites were selected from the identification of all triplicate replicates in at least one sample group. The significantly enriched phosphorylation sites were selected by the ANOVA test with a permutation-based FDR cut-off of 0.01 and S0 of 0.2. The principle component analysis (PCA) was performed using the phosphorylation sites identified across all Col-0 and *snrk2-dec* mutant plants with a cut-off of Benjamin-Hochberg FDR < 0.05. For hierarchical clustering, the intensities of the ANOVA significant phosphorylation sites were first z-scored and clustered using Euclidean as a

distance measure for row clustering. The number of clusters was set at 250, with a maximum of 10 iterations and 1 restart. The protein annotation search was performed using PANTHER database, and enrichment of cellular component was performed using Fisher's exact test with a cut-off of $p < 0.05$.

**Generating high-order mutants of OKs**. The constructs for CRISPR were designed according to the protocol described previously[36]. The sgRNAs for CRISPR/Cas9 vectors to edit RAFs are listed in Supplementary Table 1. To generate $OK^{130}$-weak, a vector pCAMBIA-1300-6RAFs containing 6 sgRNAs was used to transform raf16 (Salk_007884). A vector pCAMBIA-2300-2RAFs containing 4 sgRNAs was used to transform $OK^{130}$-weak to generate $OK^{130}$-null. To generate $OK^{100}$-quin, a vector pCAMBIA-2300-5RAFs containing 6 sgRNAs was used to transform Col-0 wild type. The fourth vector pCAMBIA-2300-11RAFs containing 8 sgRNAs was used to transform $OK^{130}$-weak to generate OK-quatdec and OK-quindec. All transgenic plants were screened for hygromycin or kanamycin resistance. The surviving T1 transformants were analyzed by sequencing their target regions, which were amplified by PCR using primer pairs listed in Supplementary Data 7.

**Electrolyte leakage assay**. To measure ion leakage in seedlings induced by PEG treatment, 5-day-old wild type, $OK^{130}$-null and OK-quindec seedlings were rinsed briefly in distilled water, and placed in a solution containing 30% PEG for 5 h. After treatment, seedlings were rinsed briefly in distilled water and placed immediately in a tube with 5 mL of water. The tube was agitated gently for 3 h before the electrolyte content was measured. Three replicates of each treatment were conducted.

**Measurement of gene expression**. Total RNA was extracted from wild type, $OK^{130}$-null and OK-quatdec seedlings treated with 300 mM mannitol and 50 μM ABA for 6 h. Total RNA was isolated using the RNeasy Plant Mini Kit (QIAGEN) according to the manufacturer's instructions. For real-time PCR assays, reactions were set up with iQ SYBR Green Supermix (Bio-Rad). A CFX96 Touch Real-Time PCR Detection System (Bio-Rad) was used to detect amplification levels. Quantification was performed using three independent biological replicates.

The expression values of the RAF genes in different tissues, and under abiotic stress and hormonal treatments were downloaded from the Arabidopsis eFP Browser "Developmental Map", "Abiotic Stress" and "Hormone" data sources, respectively (http://bar.utoronto.ca/efp/cgi-bin/efpWeb.cgi). Heatmaps were generated using the R package "pheatmap" (https://cran.r-project.org/package = pheatmap).

**Water loss measurement**. For the measurement of water loss, detached rosette leaves of 4-week-old plants were placed in weighing dishes and left on the laboratory bench with light. Fresh weight was monitored at the indicated time. Water loss was expressed as a percentage of initial fresh weight.

**Split luciferase (LUC) complementation assay**. The coding sequence of RAF35 and SnRK2s was amplified by PCR, cloned into pENTR vectors and transferred to pEarley-nLUC/cLUC vectors through LR reactions. Split-LUC complementation assay was performed by transient expression in tobacco leaves through agrobacterium-mediated infiltration. Two days after infiltration, luciferase activity was detected with a CCD camera by applying firefly D-luciferin (NanoLight).

**Protein expression, purification, and in vitro kinase assay**. cDNA fragments encoding full-length SnRK2s were cloned into pET-28a vector with the coding sequence of $6 \times$ HIS-tag, a thrombin cleavage site, and a T7 tag fused. cDNA fragments encoding kinase domains of the RAFs were cloned into pGEX-4T-1 and pMal-c2X vectors with the primer listed in Supplementary Data 7. The resulting plasmids were transformed into BL21 or ArcticExpression cells. The recombinant proteins were expressed and purified using standard protocols. For the phosphorylation assay, recombinant full-length SnRK2s and kinase domains of RAFs (aa814-1117 for RAF40-KD, aa969-1237 for RAF24-KD, aa734-1030 for RAF5-KD, aa650-933 for RAF2-KD, aa644-956 for RAF6-KD, aa466-831 for RAF10-KD, aa473-773 for RAF7-KD) were incubated in reaction buffer (25 mM Tris HCl, pH 7.4, 12 mM MgCl$_2$, 2 mM DTT), with 1 μM ATP plus 1 μCi of [γ-$^{32}$P] ATP for 30 min at 30 °C. To detect the effects of RAF phosphorylation on SnRK2.4/6 kinase activity, recombinant SnRK2.4/6 were pre-incubated with recombinant GST-ABI1 coupled on beads in 25 mM Tris-HCl, pH 7.4, 12 mM MgCl$_2$ and 2 mM DTT for 30 min at 30 °C. After a brief centrifuge to remove the ABI1, aliquots of SnRK2.4/6 were subjected to in vitro kinase reactions with or without the presence of wild type or "kinase-dead" forms of RAFs. After 30 min incubation at 30 °C, ABF2 and 1 μCi [γ-$^{32}$P] were added to the reaction and incubated for an additional 30 min at 30 °C. Reactions were terminated by boiling in SDS sample buffer and separated by 10% SDS-PAGE.

**Immunoblotting**. 30 mg samples were ground into fine powder in liquid N$_2$ and dissolved in 100 μL protein extract buffer (100 mM HEPES, pH 7.5, 5 mM EDTA, 5 mM EGTA, 10 mM DTT, 10 mM Na$_3$VO$_4$, 10 mM NaF, 50 mM β-glycerophosphate, 1 mM PMSF, 5 μg/mL leupeptin, 5 μg/mL antipain, 5 μg/mL aprotinin, and 5% glycerol) followed by centrifugation at for 40 min at 4 °C. The supernatants were separated by 12% SDS/PAGE. After electrophoresis, the proteins were transferred to PVDF membrane and immunoblotted with antibodies against SnRK2.2/3/6 and SnRK2.6-p-S175. Immunoblot with anti-actin was used as the loading control.

**Yeast two hybrid assay**. To detect protein interactions between RAFs and SnRK2s, pGADT7 plasmids containing RAFs were co-transformed with wild-type or mutated pGBKT7-SnRK2s into Saccharomyces cerevisiae AH109 cells. Successfully transformed colonies were identified on yeast SD medium lacking Leu and Trp. Colonies were transferred to selective SD medium lacking Leu, Trp, His, and in the presence of 3-Amino-1,2,4-Triazol (3-AT). To determine the intensity of protein interaction, saturated yeast cultures were diluted to $10^{-1}$, $10^{-2}$, and $10^{-3}$ and spotted onto selection medium. Photographs were taken after 4 days incubation.

**Confocal microscopy**. Seven-day-old seedlings of RAF-GFP were imaged using a Leica TCS SP8 laser scanning confocal microscope at 488 nm laser excitation and 500 to 550 nm emission for GFP.

**Quantification and statistical analysis**. Student's t-test was used to determine the statistical significance between wild type and mutants in assays related to germination, root length, fresh weight, relative intensity, or relative abundance.

**Reporting Summary**. Further information on research design is available in the Nature Research Reporting Summary linked to this article.

## Data availability
The phosphoproteomic data were deposited to the ProteomeXchange Consortium via the PRIDE partner repository with the dataset identifier PXD014435. Source data underlying Fig. 1a–b; 3c; 4c–f; 5c–f, as well as Supplementary Figs. 2a, d, f; 3a; 6a, b, d; 7a, d; 8f; 9a–d, f–g; 10 are provided as a Source Data file. Source data underlying Figs. 2a, c, d are also available in Supplementary Data 4 and 5. Other data supporting the findings of this study are available within the manuscript and its supplementary files or are available from the corresponding authors upon request.

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

## Acknowledgements
This work was supported by the Strategic Priority Research Program of the Chinese Academy of Sciences, Grant XDB27040106 (to P.W.), XDB27040000 (to J.-K.Z.), and National Natural Science Foundation of China, Grant 31771358 (to P.W.). Y.L. is supported by China Scholarship Council. We are grateful to Prof. Dingzhong Tang of Fujian Agriculture and Forestry University for providing the materials of EDR1, to Drs. Jiamu Du of the Shanghai Center for Plant Stress Biology and Huazhong Shi of the Texas Tech University for helpful discussions. We would like to thank Life Science Editors for editorial assistance and AOMICS (Shanghai, China) for assistance on proteomics analyses.

## Author Contributions
J.-K.Z. and P.W. conceived the project and designed research; Z.L., Y.L., Z.Z., X.L., C.-C.H., Y.D., T.S., W,Y., P.P., S.V., C.Z., Y.Z. and P.W. performed research; Z.L., Y.L., Z.Z., X.L., C.-C.H., T.S., S.V., C.-P.S., W.A.T., P.W. and J.-K.Z. analyzed data; and J.-K.Z. and P.W. wrote the paper with contributions from all authors.

## Competing interests
The authors declare no competing interests.
