## [Peer Review File · Nature Communications]

Reviewers' comments:

Reviewer #1 (Remarks to the Author):

SnRK2 kinases are key regulators of osmotic stress responses in plants. The paper by Lin et al. identifies Raf-like MAPKKKs/OKs as crucial upstream kinases of SnRK2s that phosphorylate and activate SnRK2s in response to osmotic stress and to a lesser degree in response to the stress hormone ABA. B4 subgroup Raf-like MAPKKK/subgroup 1 OKs phosphorylate predominantly ABA-independent SnRK2s in response to osmotic stress, while subgroup 2 and 3 OKs phosphorylate predominantly the 3 strongly ABA-inducible SnRK2s. The experiments are comprehensive and well-controlled and the results exciting. However, there are few points that the authors need to address before I can recommend the paper for publication:

1. In a similar phosphoproteomic analysis the Sussman group also identified phosphorylation of B4 Raf-like MAPKKK in response to osmotic stress (Stecker et al, 2014. *Plant Phys.* 165, 1171). Please provide the reference.
2. The authors propose that both subgroup 2 and 3 OKs phosphorylate and activate ABA-dependent SnRK2s. The ultimate evidence would be the generation of subgroup 2-only and subgroup 3-only mutants, neither of which alone would be sufficient to prevent osmotic stress-induced phosphorylation of SnRK2.2/2.3/2.6. Why have the authors only generated the combination of subgroup 2+3 mutants?
3. All the blot and gel images would benefit from better labeling of bands and inclusion of labeled size standards. That is particularly the case for Figs. 3D and 4C,D. Fig. 1 shows that the ABA-inducible SnRK2s (2.2, 2.3, 2.6) migrate more slowly on SDS PAGE gels than the ABA-independent SnRK2s, yet in Fig. 3D SnRK2.4 migrates with the same mobility as SnRK2.6 and more slowly than SnRKs2.1 and 2.10. The explanation is that for the *in vitro* kinase assays shown in Figs. 3D/4C/4D only the kinase domains were used rather than full length SnRK2s. This is not mentioned in the main text and I had to go into the Methods part to find out. The Methods part also lacks details (e.g., aa range) for the KD-only constructs. This information is important for the interpretation of results as detailed in point 4.
4. Based on their results using KD-only constructs of SnRK2s, the authors suggest that SnRK2s do not autophosphorylate, which "provides an important update on our current understanding of the ABA core signaling pathway". I find this a misleading statement as it has been shown that autophosphorylation requires the SnRK2 box in addition to the kinase domain and that mutations that break the interaction between kinase domain and SnRK2 box compromise kinase activity (e.g., Ng et al., 2011, *PNAS* 108, 21259).
5. The authors have provided convincing evidence that subclass 2/3 OKs phosphorylate and activate the ABA-dependent SnRK2s in response to osmotic stress. However, their role in activation and phosphorylation in response to ABA is more complex. While in OK100-quin mutants (mutations of 5 subclass 2/3 OKs) phosphorylation of ABA-dependent SnRK2s is almost abolished in response to osmotic stress, phosphorylation is not affected in response to ABA (Fig. 2D). Yet ABA-dependent phosphorylation is reduced in the OK-quindec mutant, pointing to a possible role of subgroup 1 OKs (in contrast to activation by osmotic stress). Interestingly, osmotic stress-induced phosphorylation of the crucial S175 in the ABA-dependent SnRK2.6 is abolished in the OK130-null mutant (mutation of all subgroup 1 OKs) (Fig. 3F; surprisingly, the authors did not test the effect of OK100 mutants on S175 phosphorylation). These are interesting results that need to be discussed.
6. If phosphorylation of activation loop S175 and S171 is responsible for the activation of SnRK2.6 in response to osmotic stress, can you speculate why mutations in PP2Cs (which dephosphorylate both residues) do not affect the osmotic stress response?

Reviewer #2 (Remarks to the Author):

This work by Lin, Li and Zhang et al. describes a group of protein kinase in plants, named OK, in osmotic stress signaling. The authors identified OKs from comparative phosphoproteomic analysis of Arabidopsis WT and *snrk2dec* mutants treated with osmotic stress. Higher-order mutants of OKs were established and used for functional analysis of OKs in osmotic stress signaling. They showed that OKs act as upstream regulators of SnRK2, which are known as major protein kinases in osmotic stress signaling. Finally, the authors proposed a model in which each subgroup of OK has specificity to each subgroup of SnRK2. Overall, experiments are well designed; and the data quality is good. Their findings will bring new insights to osmotic stress signaling in plants. However, there are some problems in this manuscript as follows.

[Major points]

1. The major conclusion in this manuscript is that B-group Raf kinases regulate SnRK2 during osmotic stress signaling in plants. The authors mentioned that their discovery brings a breakthrough in this research field. However, it seems that the authors have overstated the novelty of this work. It already has been reported that a B-group Raf kinase, ARK, acts as an upstream regulator of SnRK2 in response to ABA and osmotic stress in *Physcomitrella patens* (Saruhashi et al. PNAS 2015). Although this reviewer can understand how the authors took their original approach to find B-group Rafs using phosphoproteomic data, the significance of this work is actually that B-group Raf kinases are diverse and that each member has distinctive roles in higher plants, as compared to bryophytes. Therefore, the authors should rewrite the manuscript to put it in better context with previous work. In particular, the authors should mention the previous work related to Raf kinases in ABA or stress signaling in the Introduction.

2. The authors named B-group Raf kinases as 'Osmotic stress activated protein Kinase', OK. However, this family had been already designated Raf10, 11..., and the nomenclature has been widely accepted (e.g. the following articles):

1. Lee SJ, Lee MH, Kim JI, Kim SY. Arabidopsis putative MAP kinase kinase kinases Raf10 and Raf11 are positive regulators of seed dormancy and ABA response. *Plant Cell Physiol.* 2015 Jan;56(1):84-97. doi: 10.1093/pcp/pcu148.
2. Virk N, Li D, Tian L, Huang L, Hong Y, Li X, Zhang Y, Liu B, Zhang H, Song F. Arabidopsis Raf-Like Mitogen-Activated Protein Kinase Kinase Kinase Raf43 Is Required for Tolerance to Multiple Abiotic Stresses. *PLoS One.* 2015 Jul29;10(7):e0133975. doi: 10.1371/journal.pone.0133975.
3. Wang B, Liu G, Zhang J, Li Y, Yang H, Ren D. The RAF-like mitogen-activated protein kinase kinase kinases RAF22 and RAF28 are required for the regulation of embryogenesis in Arabidopsis. *Plant J.* 2018 Nov;96(4):734-747. doi:10.1111/tpj.14063.

Therefore, it is better to use the existing nomenclature rather than introducing a new name for this family. It will only bring confusion to the field. The authors should replace OK names with "Raf #" throughout this manuscript.

3. If I understand correctly, the authors analyzed phosphoproteomics data to extract differentially regulated phosphopeptides in response to mannitol treatment between WT and *snrk2dec*. In such a comparative analysis, authors should use quantitative data of each phosphopeptides. However, this reviewer cannot find this information in Dataset S1-S5. In addition, graphs for quantitative data of all B2, B3 and B4 Rafs should be included in supplemental data to make the information easier to understand for readers.

4. In Extended Data Figure 9, authors proposed a model in which B2 and B3 Rafs regulate subclass II and III SnRK2s, and B4 Rafs regulate subclass I SnRK2s, in response to osmotic stress. However, Raf mutants showed ABA-insensitive phenotype. Does this mean ABA can somehow affect Rafs? Furthermore, authors showed that B4 Rafs can interact with all SnRK2s. How about other Rafs? In addition, this data does not support the model presented in Extended Data Figure 9.

5. In Figure 4, expression of some stress-responsive genes were analyzed. ABA treatment should be included in this figure. It is well known that those genes are regulated by some transcription factors, i.e. ABF/AREB or CBF/DREB etc. Therefore, it would be useful to analyze expression of those transcription factors in WT and mutants.

6. No information on cellular localization of Raf kinases. For example, CTR1(OK13) is localized in ER, and it is important for its function. How about other Rafs (OKs)? Such information should be useful to discuss about functions of each Raf, especially for the relationship between Raf and SnRK2 .

7. No information on expression pattern of Raf genes, e.g. ABA/stress response, tissue specificity or developmental stages.

[Minor points]

8. In Figure 1, molecular mass of each protein should be indicated.

9. In Extended Data Figure 5 (B) legend, GFP-OK1 should be GFP-OK2?

10. L85-86, "18 and 30 bp deletions in OK3 and OK7, respectively." Is this correct? Extended Data Figure 2F showed 30 bp deletion in OK3, and 18 bp deletion in OK7.

11. L137, SnRK2.2s should be SnRK2s.

12. Authors sometimes misspells 'SnRK2' as 'SnKR2' in the manuscript.

13. In Extended Data Figure 7D, OK15/RAF10-KD should be OK16/RAF10-KD.

Reviewer #3 (Remarks to the Author):

Lin et al. present a tour de force addressing the mechanism of osmoregulation via a kinase cascade in higher plants. Using in gel kinase assays, they identified 4 size classes of kinases activated by osmotic stress. Using higher order mutants to knock out co-regulated kinases, they first identified the subset of 5 SnRK2 kinases that are activated by osmotic stress, but not ABA. Given that this was based on analysis of a septuple mutant, it was not clear how they focused on just 5 of the 7 genes knocked out in this mutant.

In addition to the 37-40 kDa kinases, osmotic stress induced 2 much larger classes of kinase that they designated OKs. Identification of the OKs made use of phosphoproteomic analyses of osmotically stressed wt vs. mutants lacking all 10 ABA- and osmotically-induced SnRKs. This identified 18 candidate kinases of the Raf-like MAPKKK class that corresponded to appropriate sizes for the 2 larger classes of osmotically activated kinases. These were again tested functionally by mutant analysis, beginning with single mutants and then using CRISPR to create higher order mutants knocking out or down 5-15 genes at a time. In addition to eliminating subsets of the larger osmotically activated kinases, these mutants had generally poor growth and were hypersensitive to osmotic stress effects on growth inhibition, ion leakage, and SnRK2.4 activation as reflected in phosphorylation of ABF2. The most severe combinations also had poor seed set, so some studies were done with slightly healthier weak mutant lines.

Direct interactions between the OKs and the SnRK2s were demonstrated by IP followed by MS, split-LUC assays, and in vitro phosphorylation assays. MS analyses identified target residues within the SnRK2s, which were then functionally tested by mutation. In contrast to wt OK kinase domains, "Kinase-dead" OK mutants were found to be ineffective in activating SnRK2s in vitro. Overall, they have identified a large set of redundant kinases acting in parallel to and converging with the "ABA core signaling pathway" to mediate stress responses through both shared and

distinct subsets of SnRK2s. A relatively novel aspect is the apparent direct activation of SnRK2s by MAPKKK homologs, without apparent need for other intermediates of a MAPK cascade. Although the osmosensor(s) regulating OK activity are still unknown, this ms. represents a major step forward in our understanding of osmoregulatory signaling in plants.

Point to point responses to Reviewer's comments:

Reviewers' comments:

Reviewer #1 (Remarks to the Author):

SnRK2 kinases are key regulators of osmotic stress responses in plants. The paper by Lin et al. identifies Raf-like MAPKKKs/OKs as crucial upstream kinases of SnRK2s that phosphorylate and activate SnRK2s in response to osmotic stress and to a lesser degree in response to the stress hormone ABA. B4 subgroup Raf-like MAPKKK/subgroup 1 OKs phosphorylate predominantly ABA-independent SnRK2s in response to osmotic stress, while subgroup 2 and 3 OKs phosphorylate predominantly the 3 strongly ABA-inducible SnRK2s. The experiments are comprehensive and well-controlled and the results exciting. However, there are few points that the authors need to address before I can recommend the paper for publication:

1. In a similar phosphoproteomic analysis the Sussman group also identified phosphorylation of B4 Raf-like MAPKKK in response to osmotic stress (Stecker et al, 2014. Plant Phys. 165, 1171). Please provide the reference.

Response: We have revised the **Introduction** and added the reference (Stecker et al., 2014), also citing additional references about Raf-like kinases and stress signaling.

2. The authors propose that both subgroup 2 and 3 OKs phosphorylate and activate ABA-dependent SnRK2s. The ultimate evidence would be the generation of subgroup 2-only and subgroup 3-only mutants, neither of which alone would be sufficient to prevent osmotic stress-induced phosphorylation of SnRK2.2/2.3/2.6. Why have the authors only generated the combination of subgroup 2+3 mutants?

Response: We thank the reviewer for the constructive suggestion. In our in-gel kinase assay result, we found that the ABA-induced activation of SnRK2s is strongly impaired in the *OK-quindec* mutant, but not in the *OK¹³⁰-null* allele. OKs from subgroups II/III phosphorylated and activated SnRK2.6 *in vitro*. Thus, we propose that both subgroup-II and subgroup-III OKs are involved in ABA signaling. We agree with the reviewer that it is important to dissect the role of each subgroup of OKs in ABA-induced activation of SnRK2.2/3/6. We are currently generating the subgroup II-only and subgroup III-only mutants by introducing CRISPR-Cas9 constructions in WT background. However, it will take at least one year to get the homozygous lines carrying knock-out mutations in six genes in each subgroup. Thus, we are unable to include this result in the current manuscript.

3. All the blot and gel images would benefit from better labeling of bands and inclusion of labeled size standards. That is particularly the case for Figs. 3D and 4C,D. Fig. 1 shows that the ABA-inducible SnRK2s (2.2, 2.3, 2.6) migrate more slowly on SDS PAGE gels than the ABA-independent SnRK2s, yet in Fig. 3D SnRK2.4 migrates with the same mobility as SnRK2.6 and more slowly than SnRK2.1 and 2.10. The explanation is that for the in vitro kinase assays shown in Figs. 3D/4C/4D only the kinase domains were used rather than full length SnRK2s. This is not mentioned in the main text and I had to go into the Methods part to find out. The Methods part also lacks details (e.g., aa range) for the KD-only constructs. This information is important for the interpretation of results as detailed in point 4.

Response: We thank the reviewer for the comment. We have now added labeled size standards to the blot and gel images in the revised Figures and Supplementary Figures. We agree with the reviewer that the mobilities of SnRK2 proteins differ from their molecular weights. For example, SnRK2.6 (41.0 kDa), SnRK2.2 (42.0 kDa) SnRK2.3 (41.1 kDa) are similar to or smaller than SnRK2.4 (42.1 kDa) and SnRK2.10 (41.0 kDa); however, for unknown reasons, SnRK2.6/2.2/2.3 migrate more slowly than SnRK2.4/2.10 and other ABA-independent SnRK2s in the in-gel kinase assay (Fig. 1a, see also references from Boudsocq, et al., 2004; Fujii et al., 2007; Zhao et al., 2018).

We apologize if any description in our text caused misunderstanding. In all the *in vitro* kinase assays, we used the **full-length SnRK2s**, not the kinase domain-only versions. We have added the detailed aa ranges of OK-KD in the revision.

(Reviewer 1 Comment 4) *4. Based on their results using KD-only constructs of SnRK2s, the authors suggest that SnRK2s do not autophosphorylate, which "provides an important update on our current understanding of the ABA core signaling pathway". I find this a misleading statement as it has been shown that autophosphorylation requires the SnRK2 box in addition to the kinase domain and that mutations that break the interaction between kinase domain and SnRK2 box compromise kinase activity (e.g., Ng et al., 2011, PNAS 108, 21259).*

Response: As mentioned in response to **Reviewer 1 comment 3**, we used the **full-length SnRK2s**, not the kinase domain-only versions in the all the kinase assays. As far as possible, we followed the methods of Ng et al., 2011, *PNAS* and Soon et al., 2012, *Science*, to perform the protein purification and *in vitro* kinase assays. Nonetheless, we did not detect the self-activation activity of either dephosphorylated SnRK2.4 or dephosphorylated SnRK2.6. Instead, our results showed that OKs are necessary to activate dephosphorylated SnRK2s *in vitro* and are essential for SnRK2 activation *in vivo*. Based on our results and several independent studies reported at conferences/symposia, we conclude that the phosphorylation of SnRK2s by OKs is necessary for SnRK2 activation. We added discussion on this point in the revision.

(Reviewer 1 Comment 5) 5. The authors have provided convincing evidence that subclass 2/3 OKs phosphorylate and activate the ABA-dependent SnRK2s in response to osmotic stress. However, their role in activation and phosphorylation in response to ABA is more complex. While in OK100-quin mutants (mutations of 5 subclass 2/3 ODs) phosphorylation of ABA-dependent SnRK2s is almost abolished in response to osmotic stress, phosphorylation is not affected in response to ABA (Fig. 2D). Yet ABA-dependent phosphorylation is reduced in the OK-quindec mutant, pointing to a possible role of subgroup 1 OKs (in contrast to activation by osmotic stress). Interestingly, osmotic stress-induced phosphorylation of the crucial S175 in the ABA-dependent SnRK2.6 is abolished in the OK130-null mutant (mutation of all subgroup 1 OKs) (Fig. 3F; surprisingly, the authors did not test the effect of OK100 mutants on S175 phosphorylation). These are interesting results that need to be discussed.

Response: We thank the reviewer for the constructive suggestion. We have added some discussion in the revision to discuss the role of subgroup II/III OKs in ABA signaling.

We also added a result showing that ABA-induced S175 phosphorylation is also impaired in the *OK-quindec*, but not in the *OK¹³⁰-null* or *OK¹³⁰-weak*, when compared to the wild type (Supplementary Fig. 9f).

(Reviewer 1 Comment 6) 6. If phosphorylation of activation loop S175 and S171 is responsible for the activation of SnRK2.6 in response to osmotic stress, can you speculate why mutations in PP2Cs (which dephosphorylate both residues) do not affect the osmotic stress response?

Response: We thank the reviewer for the comment. From our mass-spec results (Table S6), we know that besides PP2C target sites S175 and S171, OKs also phosphorylate other sites at both the N- and C-terminus of SnRK2.6. Some additional phosphorylation on phosphosites in the ABA box or near the activation loop may disrupt the interaction between SnRK2 and PP2C. Our preliminary yeast-three hybrid assay result supports the notion that adding OKs reduces the interaction between ABI1 and SnRK2.6. We have added some discussion in the text but would like to keep these results until we have more solid structural/biochemical evidence and a clear mechanism.

Reviewer #2 (Remarks to the Author):

This work by Lin, Li and Zhang et al. describes a group of protein kinase in plants, named OK, in osmotic stress signaling. The authors identified OKs from comparative phosphoproteomic analysis of Arabidopsis WT and snrk2dec mutants treated with osmotic stress. Higher-order mutants of OKs were established and used for functional analysis of OKs in osmotic stress signaling. They showed that OKs act as upstream regulators of SnRK2, which are known as major protein kinases in osmotic stress signaling. Finally, the authors proposed a model in which each subgroup of OK has specificity to each subgroup of SnRK2. Overall, experiments are well designed; and the data quality is good. Their findings will bring new insights to osmotic stress signaling in plants. However, there are some problems in this manuscript as follows.

[Major points]

(Reviewer 2 Comment 1) 1. The major conclusion in this manuscript is that B-group Raf kinases regulate SnRK2 during osmotic stress signaling in plants. The authors mentioned that their discovery brings a breakthrough in this research field. However, it seems that the authors have overstated the novelty of this work. It already has been reported that a B-group Raf kinase, ARK, acts as an upstream regulator of SnRK2 in response to ABA and osmotic stress in *Physcomitrella patens* (Saruhashi et al. *PNAS* 2015). Although this reviewer can understand how the authors took their original approach to find B-group Rafs using phosphoproteomic data, the significance of this work is actually that B-group Raf kinases are diverse and that each member has distinctive roles in higher plants, as compared to bryophytes. Therefore, the authors should rewrite the manuscript to put it in better context with previous work. In particular, the authors should mention the previous work related to Raf kinases in ABA or stress signaling in the Introduction.

Response: We thank the reviewer for the comment. We have rewritten the introduction and cite the ARK work as well as other studies on OK functions in ABA and osmotic stress signaling.

(Reviewer 2 Comment 2) 2. *The authors named B-group Raf kinases as ‘Osmotic stress activated protein Kinase’, OK. However, this family had been already designated Raf10, 11..., and the nomenclature has been widely accepted (e.g. the following articles):*

1. Lee SJ, Lee MH, Kim JI, Kim SY. Arabidopsis putative MAP kinase kinase kinases Raf10 and Raf11 are positive regulators of seed dormancy and ABA response. *Plant Cell Physiol.* 2015 Jan;56(1):84-97. doi: 10.1093/pcp/pcu148.
2. Virk N, Li D, Tian L, Huang L, Hong Y, Li X, Zhang Y, Liu B, Zhang H, Song F. Arabidopsis Raf-Like Mitogen-Activated Protein Kinase Kinase Gene Raf43 Is Required for Tolerance to Multiple Abiotic Stresses. *PLoS One.* 2015 Jul29;10(7):e0133975. doi: 10.1371/journal.pone.0133975.
3. Wang B, Liu G, Zhang J, Li Y, Yang H, Ren D. The RAF-like mitogen-activated protein kinase kinase kinases RAF22 and RAF28 are required for the regulation of embryogenesis in Arabidopsis. *Plant J.* 2018 Nov;96(4):734-747. doi:10.1111/tpj.14063.

Therefore, it is better to use the existing nomenclature rather than introducing a new name for this family. It will only bring confusion to the field. The authors should replace OK names with “Raf #” throughout this manuscript.

Response: We understand the reviewer's concerns and we thank the reviewer for the thoughtful suggestion. We chose to use OKs instead of Rafs for three reasons. 1. Although OKs are classed as Raf-B like MAP Kinase Kinase Kinases based on sequence similarity, our results show that rather than phosphorylating MPKKs, OKs directly phosphorylate SnRK2s and mediate SnRK2 activation upon osmotic stress. Whether OKs can act as functional MAPKKs to phosphorylate MPKKs is still unknown. As SnRK2s only exist in the plant kingdom, we propose that some OKs have specialized functions in plants, which may differ from Raf protein kinases in animals; 2. We are not sure if Rao et al (DNA Research, 2010) was the original literature of Raf nomenclature, but it is the only paper we found that mentions all Raf proteins. Based on the Raf nomenclature, the seven very close homologues of the Raf B4 subfamily, OK1 to OK7, were named RAF16, RAF40, RAF24, RAF18, RAF20, RAF35 and RAF42, which may mislead readers. Besides the three references listed by the reviewer, we found only one more paper using Raf nomenclature in the PubMed database:

Hwang JU, Yim S, Do THT, Kang J, Lee Y. Arabidopsis thaliana Raf22 protein kinase maintains growth capacity during postgerminative growth arrest under stress. *Plant Cell Environ.* 2018 Jul;41(7):1565-1578. doi: 10.1111/pce.13199

3. The GeneBank and the widely used Arabidopsis databases, TAIR and ARAPORT, also do not use the Raf nomenclature. We cannot find any genes matching RAF16, RAF40, RAF24, RAF18, RAF20, RAF35 and RAF42 from these databases.

Based on these reasons, especially the specialized function of OKs in plants, we would prefer to keep the OK names rather than changing them to Rafs.

(Reviewer 2 Comment 3) 3. *If I understand correctly, the authors analyzed phosphoproteomics data to extract differentially regulated phosphopeptides in response to mannitol treatment between WT and snrk2dec. In such a comparative analysis, authors should use quantitative data of each phosphopeptides. However, this reviewer cannot find this information in Dataset S1-S5. In addition, graphs for quantitative data of all B2, B3 and B4 Rafs should be included in supplemental data to make the information easier to understand for readers.*

Response: We thank the reviewer for the comment. The quantitative data of all kinases that respond to osmotic stress are presented in Datasets S1 to S5. Datasets S1 and S2 include all the quantifiable phosphosites in wild type and *snrk2-dec* mutant, respectively. The fold change and *t*-test significance of each phosphosite can be found in columns J/K and V, respectively. To make the data easier to read, all phosphosites from all protein kinases are listed separately as Dataset S3. The mannitol-induced phosphosites in wild type and *snrk2-dec* mutants are also listed separately as Datasets S4 and S5.

Following the reviewer's comment, the graphs for quantitative data of some mannitol-induced phosphosites in subgroup I and II/III OKs are now presented as Fig. 2c and 2d in the revision, respectively.

(Reviewer 2 Comment 4) 4. *In Extended Data Figure 9, authors proposed a model in which B2 and B3 Rafs regulate subclass II and III SnRK2s, and B4 Rafs regulate subclass I SnRK2s, in response to osmotic stress. However, Raf mutants showed ABA-insensitive phenotype. Does this mean ABA can somehow affect Rafs?*

Response: We thank the reviewer for the comment. We found that *ok-quandec* shows ABA-insensitive phenotype and subclass II and III OKs phosphorylate SnRK2.6 directly. These data suggest that OKs are also required for ABA-induced activation of SnRK2.2/3/6. As mentioned in our response to **Reviewer 1 Comment 4**, we are generating higher order mutants of subgroup II and subgroup III OKs, which will hopefully further dissect the function of OKs in ABA signaling. Whether and how ABA may affect Rafs are unclear at this time, although we did notice that the OKs appear slightly activated at early stages of ABA treatment (Fig. 1a, highlighted below), even though the activation is much weaker than that upon mannitol treatment.

(Reviewer 2 Comment 5) Furthermore, authors showed that B4 Rafs can interact with all SnRK2s. How about other Rafs? In addition, this data does not support the model presented in Extended Data Figure 9.

Response: In revised Supplementary Fig 9e, we have provided a Y2H assay result showing that subgroup II/III OKs interact with SnRK2.6 but not SnRK2.4, which supports our in-gel kinase assay and *in vitro* kinase assay results that subgroup II/III OKs prefer to interact with and phosphorylate ABA-activated SnRK2s.

Although our IP-MS and split LUC data (Supplementary Fig. 7b, c) show that subgroup I OKs (B4 Raf-like) can associate with not only ABA-independent SnRK2s but also ABA-dependent SnRK2s, it is unclear which types of SnRK2s interact more strongly with subgroup I OKs in planta. More importantly, our *in vitro* kinase assays show clearly that subgroup I OKs prefer to phosphorylate the ABA-independent SnRK2s (Fig. 4d and Supplementary Fig. 7d) and the in-gel kinase assay results with the different mutants provide strong genetic evidence supporting this preference. Therefore, the IP-MS and split LUC data showing possible physical interactions should be taken as evidence to contradict the conclusions based on strong biochemical and genetic data (i.e. the subgroup I OKs may not phosphorylate the ABA-dependent SnRK2s even if they are together physically).

(Reviewer 2 Comment 6) 5. In Figure 4, expression of some stress-responsive genes were analyzed. ABA treatment should be included in this figure. It is well known that those genes are regulated by some transcription factors, i.e. ABF/AREB or CBF/DREB etc. Therefore, it would be useful to analyze expression of those transcription factors in WT and mutants.

Response: Following the reviewer's suggestion, we have provided the transcript data of ABA treatment in the revised Fig. 5f. We also checked the expression of ABA-responsive transcription factors (Supplementary Fig. 9g in the revision).

(Reviewer 2 Comment 7) 6. No information on cellular localization of Raf kinases. For example, CTR1(OK13) is localized in ER, and it is important for its function. How about other Rafs (OKs)? Such information should be useful to discuss about functions of each Raf, especially for the relationship between Raf and SnRK2.

Response: We thank the reviewer for the good suggestion. In revised Supplementary Fig. 5, we have added localization data for GFP-OK5, GFP-OK14, GFP-OK15/RAF11, and GFP-OK19, from GFP-fusion transgenic plants we have in hands. These data show that these OKs are localized in the cytosol and some tiny spots, which might be p-bodies. As SnRK2s are known to localize in cytosol, nucleus, and p-body (Soma et al., 2017, Nature Plants), together with our split-LUC assay/IP-MS assay results, these data support that SnRK2s and the OKs may co-localize.

(Reviewer 2 Comment 8) 7. No information on expression pattern of Raf genes, e.g. ABA/stress response, tissue specificity or developmental stages.

Response: We thank the reviewer for the good suggestion. In revised Supplementary Fig. 4b-e, we have used heatmaps to show the expression of OKs in different tissues, stress treatments and hormone applications. These data were obtained from Arabidopsis eFP browser (<http://bar.utoronto.ca/efp/cgi-bin/efpWeb.cgi>). The description of these data was also added in the revision.

[Minor points]

(Reviewer 2 Comment 9) 8. In Figure 1, molecular mass of each protein should be indicated.

Response: The molecular mass of size standards was added to Fig. 1 as well as other Figures in the revision.

(Reviewer 2 Comment 10) 9. In Extended Data Figure 5 (B) legend, GFP-OK1 should be GFP-OK2?

Response: We apologize for the typo. It has been corrected in the revision.

(Reviewer 2 Comment 11) 10. L85-86, "18 and 30 bp deletions in OK3 and OK7, respectively." Is this correct? Extended Data Figure 2F showed 30 bp deletion in OK3, and 18 bp deletion in OK7.

Response: We thank the reviewer for careful reading of our manuscript. We apologize for the typo and have corrected it in the revision.

(Reviewer 2 Comment 12) 11. L137, SnRK2.2s should be SnRK2s.

Response: We thank the reviewer for careful reading of our manuscript. It has been corrected in the revision.

(Reviewer 2 Comment 13) 12. Authors sometimes misspells 'SnRK2' as 'SnKR2' in the manuscript.

Response: We apologize for the typo. We have double checked the manuscript and all instances of "SnKR2" have been changed to "SnRK2".

(Reviewer 2 Comment 14) 13. In Extended Data Figure 7D, OK15/RAF10-KD should be OK16/RAF10-KD.

Response: We apologize for the typo. It has been corrected in the revision.

Reviewer #3 (Remarks to the Author):

Lin et al. present a tour de force addressing the mechanism of osmoregulation via a kinase cascade in higher plants. Using in gel kinase assays, they identified 4 size classes of kinases activated by osmotic stress. Using higher order mutants to knock out co-regulated kinases, they first identified the subset of 5 SnRK2 kinases that are activated by osmotic stress, but not ABA. Given that this was based on analysis of a septuple mutant, it was not clear how they focused on just 5 of the 7 genes knocked out in this mutant.

Response: As only the OK^{130} -null, but not the OK^{130} -weak has the osmotic stress-related phenotype, we proposed that subgroup I OKs have redundant function in regulation of ABA-independent SnRK2-activation. However, knock-out of 5 OKs in OK^{130} -weak already strongly affects the SnRK2 activation based on our in-gel kinase assay.

In addition to the 37-40 kDa kinases, osmotic stress induced 2 much larger classes of kinase that they designated OKs. Identification of the OKs made use of phosphoproteomic analyses of osmotically stressed wt vs. mutants lacking all 10 ABA- and osmotically-induced SnRK2s. This identified 18 candidate kinases of the Raf-like MAPKKK class that corresponded to appropriate sizes for the 2 larger classes of osmotically activated kinases. These were again tested functionally by mutant analysis, beginning with single mutants and then using CRISPR to create higher order mutants knocking out or down 5-15 genes at a time. In addition to eliminating subsets of the larger osmotically activated kinases, these mutants had generally poor growth and were hypersensitive to osmotic stress effects on growth inhibition, ion leakage, and SnRK2.4 activation as reflected in phosphorylation of ABF2. The most severe combinations also had poor seed set, so some studies were done with slightly healthier weak mutant lines.

Direct interactions between the OKs and the SnRK2s were demonstrated by IP followed by MS, split-LUC assays, and in vitro phosphorylation assays. MS analyses identified target residues within the SnRK2s, which were then functionally tested by mutation. In contrast to wt OK kinase domains, "Kinase-dead" OK mutants were found to be ineffective in activating SnRK2s in vitro.

Overall, they have identified a large set of redundant kinases acting in parallel to and converging with the "ABA core signaling pathway" to mediate stress responses through both shared and distinct subsets of SnRK2s. A relatively novel aspect is the apparent direct activation of SnRK2s by MAPKKK homologs, without apparent need for

other intermediates of a MAPK cascade. Although the osmosensor(s) regulating OK activity are still unknown, this ms. represents a major step forward in our understanding of osmoregulatory signaling in plants.

REVIEWERS' COMMENTS:

Reviewer #1 (Remarks to the Author):

I recommend the revised manuscript for publication.

A very minor point: the authors may want to provide details for the ~10 kD larger size of their recombinant, kinase dead SnRK2s compared to endogenous SnRK2s (I assume parts of purification tags/protease cleavage site, but couldn't find any details).

Reviewer #2 (Remarks to the Author):

This version of manuscript was improved significantly by answering the most of reviewer's comments.

Point-by-point response to referee comments

Reviewer #1 (Remarks to the Author):

I recommend the revised manuscript for publication.

A very minor point: the authors may want to provide details for the ~10 kD larger size of their recombinant, kinase dead SnRK2s compared to endogenous SnRK2s (I assume parts of purification tags/protease cleavage site, but couldn't find any details).

Response: We thank the reviewer for the suggestion. The cDNA fragments of full length SnRK2s were cloned into the original pET28a vector. The pET28a vector contains a 6xHIS-tag, a thrombin cleavage site, and a T7 tag, which result in an addition of 33 amino acid residue (about 3.6 kD) fragment to the N-termini of SnRK2.4 and SnRK2.6. This might be the reason why the recombinant SnRK2.4 and SnRK2.6 have a larger size. We have added the details in the revised method.

Reviewer #2 (Remarks to the Author):

This version of manuscript was improved significantly by answering the most of reviewer's comments.